# Associations between Coronavirus Crisis Perception, Perceived Economic Risk of Coronavirus, General Self-Efficacy, and Coronavirus Anxiety at the Start of the Pandemic: Differences by Gender and Race

**DOI:** 10.3390/ijerph19052872

**Published:** 2022-03-01

**Authors:** Samantha Garcia, Suellen Hopfer, Elouise Botes, Samuel Greiff

**Affiliations:** 1Program in Public Health, Department of Health, Society & Behavior, Susan and Henry Samueli College of Health Sciences, University of California-Irvine, Irvine, CA 92697, USA; shopfer@hs.uci.edu; 2Department of Developmental and Educational Psychology, University of Vienna, 1010 Vienna, Austria; elouise.botes@univie.ac.at; 3Department of Behavioral and Cognitive Sciences, University of Luxembourg, 4366 Esch-sur-Alzette, Luxembourg; samuel.greiff@uni.lu

**Keywords:** anxiety, pandemic, self-efficacy, gender, race, disparities

## Abstract

The coronavirus pandemic has escalated rates of anxiety in the general U.S. population. Understanding how factors associated with coronavirus anxiety at the start of the pandemic differed among populations hardest impacted by coronavirus anxiety is key to effectively remediating negatively associated health outcomes and to better understand how to address concerns of the public at the start of a global pandemic. This study was a secondary analysis of data from a cross-sectional online survey of 1165 Prolific users between 13 and 15 March 2020. Data were collected from a stratified sample of U.S. adults aged 20 or older and currently living in the United States. The sample was stratified for age, gender, and race. Coronavirus anxiety was assessed as the dependent variable, alongside three independent variables: coronavirus crisis perception, perceived economic risk of coronavirus, and general self-efficacy. Multiple linear regression assessed the associations between the independent variables and coronavirus anxiety. Interactions between independent variables and two sociodemographic variables (i.e., gender, race) were also explored. The models were adjusted for age, gender, race, education, employment, and income. The average age of participants was 45.6 ± 15.7. The majority (76%) identified as White, approximately half identified as female and reported obtaining a bachelor’s degree or higher. Coronavirus crisis perception and perceived economic risk of coronavirus were positively associated with coronavirus anxiety (β = 0.46, 95% CI = 0.41, 1.00; β = 0.14, 95% CI = 0.09, 1.00, respectively). General self-efficacy was negatively associated with coronavirus anxiety (β = −0.15, 95% CI = −1.00, −0.11). Gender and race both moderated the association between coronavirus crisis perception and anxiety. Race moderated the association between perceived economic risk and coronavirus crisis perception. These results provide a foundation to further explore cognitive factors in subgroups disproportionately affected by anxiety during the pandemic.

## 1. Introduction

In December 2019, the coronavirus identified as SARS-CoV-2 prompted a global pandemic as many fell ill to COVID-19 across the globe. By early 2020, increasing COVID-19 rates in the United States affected employment rates, social interactions, and social norms across the nation. Unsurprisingly, increased anxiety, stress, fear, and depression were more frequently observed during the pandemic compared to months prior [1,2]. On 11 March 2020, the World Health Organization declared COVID-19 a pandemic followed by U.S. President Donald J. Trump declaring a nationwide emergency on 13 March 2020 [3]. Two days later, on 15 March 2020, the U.S. began a shutdown, closing schools and businesses in an attempt to mitigate the spread of COVID-19 [3]. This unique pandemic is a new type of stressor and traumatic event which led to an increase in demand for psychiatric services [4] and identification of new symptoms in those dealing with COVID-19-related stress, anxiety and grief [5]. A U.S.-based longitudinal study found an increase in anxiety and a decrease in tiredness, calmness, happiness and optimism from early February 2020 to mid-March 2020 [6]. Anxiety disorders are among the most prevalent mental health conditions affecting the U.S. population [7]. An excessive amount of anxiety can negatively affect everyday tasks and become debilitating [7]. Stressful events can further induce anxiety and interfere with how individuals participate in daily activities and personal relationships. Studies exploring factors associated with coronavirus anxiety at the start of the pandemic provide necessary insight on how to better address anxieties stemming from the pandemic with appropriate public health messaging and interventions.

Past work has primarily explored demographic factors and pre-existing health diagnoses associated with increased anxiety during pandemics [7,8,9]. Preliminary work has begun to outline the associations between perceived risk, self-efficacy and anxiety during the pandemic [10,11]. A China-based study conducted in February 2020, found 5-min/day exposure to COVID-19 news increased odds of depression and/or anxiety [12]. Still, more work is needed to better understand which aspects of perceived risk and self-efficacy are related to anxiety, especially among individuals in the U.S. with higher rates of anxiety during the pandemic [13]. A meta-analysis found prevalence of anxiety to be approximately 35% for the general U.S. population without coronavirus and nearly 32% among patients with coronavirus [13]. Future studies should elucidate on how the impact of perceived risk and self-efficacy on coronavirus anxiety may differ among individuals most affected by anxiety during the pandemic. Studies have consistently found women to be among subgroups most adversely affected by anxiety throughout the coronavirus pandemic [13,14,15]. Some studies have found non-White individuals reported higher levels of anxiety during the pandemic [16,17], although the literature is not conclusive [18]. To further explore anxiety disparities during the coronavirus pandemic, this study explores the potential moderation of gender and race between cognitive factors (i.e., coronavirus crisis perception, perceived economic risk of coronavirus, general self-efficacy) and coronavirus anxiety.

In selected forms of anxiety, an individual may have elevated perceptions of risk while underutilizing coping mechanisms [19]. Risk perception attitude (RPA) framework posits that perceived risks are better understood within the context of efficacy beliefs [20,21]. In RPA framework, perceived risk and efficacy have both been shown to influence health behaviors and outcomes [20]. This study uses RPA framework to simultaneously evaluate the associations of perceived risk of coronavirus, perceived economic risk of coronavirus and general self-efficacy on coronavirus anxiety. Self-efficacy and perceived risk may vary by gender and race [22,23]. Understanding which aspects of perceived risk and self-efficacy impact populations most adversely affected by anxiety may better inform health messages and anxiety prevention efforts.

An important cognitive factor to understand in the context of coronavirus anxiety may be perception of the crisis. Perception of the health impact of coronavirus was associated with higher anxiety among people with high-risk type 2 diabetes [24]. Risk perception has been defined and conceptualized in many ways [25]. This study evaluated crisis perception as an individual’s view of the health threat of the coronavirus to the larger global community. Crisis perception of the pandemic’s widespread reach and impact on the global community is likely to contribute to anxiety in the general U.S. population as it has with higher-risk groups.

Many stressors may contribute to anxiety symptoms. For example, financial and economic hardship has been associated with anxiety and stress [26]. In April 2020, at the time of the study, the seasonally adjusted U.S. civilian unemployment rate rose to 14.7% [27]. The relationship between perceived economic risk of coronavirus and coronavirus anxiety may provide clarity on important factors associated with anxiety among adversely impacted populations and related exacerbated chronic health outcomes.

Self-efficacy is another cognitive factor that may be associated with coronavirus anxiety. The association between lower self-efficacy and increased general anxiety symptoms has been primarily observed among samples of frontline health workers in Wuhan, China, where the pandemic began [28,29]. It is unclear if factors such as gender and race also moderate the association between general self-efficacy and coronavirus-specific anxiety.

The present study explored cognitive factors related to coronavirus anxiety potentially modifiable through public health messaging or future interventions. The hypotheses were: (1) Higher coronavirus crisis perception will be associated with higher coronavirus anxiety; (2) Higher economic risk perception of coronavirus will be associated with higher coronavirus anxiety; and (3) Lower general self-efficacy will be associated with higher coronavirus anxiety. This study further assessed how these associations may have differed among populations at higher risk of coronavirus anxiety (e.g., females and racial/ethnic minorities). Additional hypotheses were: (1) Female gender moderates the associations between coronavirus crisis perception, economic risk perception, general self-efficacy and coronavirus anxiety; and (2) White racial status moderates the associations between coronavirus crisis perception, economic risk perception, general self-efficacy and coronavirus anxiety. For the purposes of this study, White racial status refers to non-Hispanic White individuals.

## 2. Materials and Methods

### 2.1. Study Sample

From 13 to 15 March 2020, online surveys were collected from the Prolific platform. Eligibility criteria included adults who were 20 years old or older, living in the United States, identified as U.S. citizens, and registered on Prolific. The sample was stratified by age, gender, and race. Data from the parent study are available via the online OSF platform [30]. The primary study was approved by the ethics review panel at The Université of Luxembourg [31]. The online survey was compatible with phone and computer devices. Eligibility was confirmed using Prolific participant registration characteristics and a pre-survey. Of 1339 participant attempts at opening the survey, 122 did not advance to informed consent, 31 did not fully compete the survey, 17 had a missing income value and four had tested positive for COVID-19; these participants were excluded. A sample of 1165 was obtained for the parent study.

The dataset featured more than 50 variables measured by a total of 378 unique items. The dataset was planned from the offset to be utilized for numerous hypotheses in international collaborations examining COVID-19 in the context of behavioral psychology [32], religiosity [33], and risk perception [31]. Each resultant study from the dataset was carefully planned so as to ensure that there were no overlapping aims or hypotheses. As such, due to the dataset being used for complex analyses in multiple studies, the sample size recruited was a large as possible given the financial resources made available to the researchers. The large sample can be considered especially beneficial to this study as analyses can be considered stable and not under-powered [34]. In addition, due to the stratification capabilities of Prolific, the study can be considered a fair representation of the age, gender, and race distributions of the U.S. population [35]. The current study is therefore a secondary data analysis using the same sample (*N* = 1165). Analyses were derived from a dataset used to answer other research questions [31], but research questions in this study were unique.

### 2.2. Measures

A 10-item coronavirus anxiety scale [33] (The Coronavirus Anxiety Scale was used as a two-dimensional scale in previous publications. In this study, the Coronavirus Anxiety Scale was applied as a one-dimensional scale because of its high internal validity and consistent relationship between each of the two dimensions and the independent variables explored in this study), was the dependent variable informed by the cognitive–somatic anxiety framework [36,37]. Answers to 5 cognitive and 5 somatic items were summed and averaged with internal consistency (α = 0.85). Higher scores reflected higher anxiety. Example survey items are: “I do not think that coronavirus will affect my life” (reverse coded; cognitive) and “My heart beats faster when I think about catching coronavirus” (somatic). A 5-point Likert response format ranging from strongly disagree to strongly agree was used.

Coronavirus crisis perception, perceived economic risk of coronavirus, and general self-efficacy comprised the independent variables for the analysis. The 5-item Coronavirus Crisis Perception Scale related to perceptions and attitudes regarding the pandemic as a global crisis and was developed for the study (α = 0.82). Responses were summed and averaged. Higher scores reflected higher coronavirus crisis perception. Example items are: “I think the world is facing an unprecedented health emergency because of coronavirus” and “I think coronavirus is not a serious threat to the world” (reverse coded). A 1-item measure for perceived economic risk of the coronavirus was used: “I think coronavirus will be a disaster for our economy”. A higher score indicated higher perception of risk that the coronavirus poses for the U.S. economy. An 8-item self-efficacy measure was adapted from the New General Self-Efficacy Scale [38]. Responses were summed and averaged, with internal consistency of α = 0.91. Higher scores reflect higher general self-efficacy. Example items are: “I will be able to successfully overcome many challenges” and “Even when things are tough, I can perform quite well”. All measures used a 5-point Likert scale with responses ranging from strongly disagree to strongly agree.

Six covariates (age, gender, race, education, employment, and income) were used in the analyses. Age in years was self-reported by participants. Participants were asked to indicate their gender as (1) female, (2) male, or (3) other. Responses were then dummy-coded for analysis. Race and ethnic data were collected as (1) Asian, (2) African American, (3) Hispanic, (4) Native American or Native Hawaiian, (5) White, or (6) other. For this specific study, race categories were recoded as (0) other and (1) non-Hispanic White. Education level was collected categorically as (1) less than a high school diploma, (2) high school diploma or equivalent (e.g., GED), (3) some college, no degree, (4) associate degree, (5) bachelor’s degree, (6) master’s degree, (7) professional degree, (8) doctoral degree, (9) vocational training or trade, and (10) other school, with a text space to specify. For this study, responses regarding educational status were recoded into (0) associate degree or less and (1) bachelor’s degree or higher. Employment status was collected as (1) employed full time (40 or more hours a week), (2) employed part time (up to 39 h a week), (3) unemployed and currently looking for work, (4) unemployed and currently not looking for work, (5) student, (6) retired, (7) homemaker, (8) self-employed, and (9) unable to work. For this specific study, responses were recoded as (1) employed full time (40 or more hours a week and (0) other. Lastly, annual gross household income was collected as (1) I do not have personal income, (2) less than USD 20,000, (3) USD 20,000 to USD 34,999, (4) USD 35,000 to USD 49,999, (5) USD 50,000 to USD 74,999, (6) USD 75,000 to USD 99,999, (7) USD 100,000 up to USD 114,999, (8) USD 115,000 up to USD 129,999, (9) USD 130,000 or more, and (10) I do not wish to respond. Gross household income responses were treated as continuous. “I do not wish to respond” was treated as missing. All measures for this study are available via the OSF platform [39].

### 2.3. Statistical Analysis

Analyses were conducted using Stata software (Version 15.1). Preliminary data screening checked data for outliers beyond three standard deviations, skewness, and kurtosis. Outliers observed in the distribution of continuous variables were not excluded because they were all within a feasible range. Robust standard errors were used to address deviations from normality. The income variable contained <2% of missing data. Case-wise deletion was used for missing observations since total missing is below the recommendation for imputation [40]. Variance inflation factor and tolerance values were computed to verify no signs of potential multicollinearity between all variables in the model. Descriptive statistics were calculated with means and standard deviations of participant characteristics. Associations between independent variables and covariates were tested using analysis of variance and simple linear regression to provide better understanding of bivariate relationships. A two-tailed *p*-value of 0.05 or less was considered significant for bivariate tests.

To test the three initial hypotheses, one multiple linear regression model assessed the associations between anxiety and the three cognitive factors, adjusting for covariates of age, gender, race, education level, employment status, and income. To test directional hypotheses, one-tailed *p*-values of 0.05 or less were considered significant for all tests. Standardized coefficients are reported. Model fit was assessed using the mean-adjusted *R*-square value. The computed multiple regression model is as follows.
$${\hat{Y}}_{i}=\beta_{0}\;+\;\beta_{1}\;(\mathrm{coronavirus}\;\mathrm{crisis}\;\mathrm{perception})\;+\beta_{2}\;(\mathrm{perceived}\;\mathrm{economic}\;\mathrm{risk}\;\mathrm{of}\;\mathrm{coronavirus})\;+\beta_{3}\;(\mathrm{general}\;\text{self-efficacy})\;+\beta_{4}\;(\mathrm{age})\;+\beta_{5}\;(\mathrm{female})\;+\beta_{6}\;(\mathrm{other}\;\mathrm{gender})\;+\beta_{7}\;(\mathrm{White})\;+\beta_{8}\;(\mathrm{bachelor}’s\;\mathrm{degree}\;\mathrm{of}\;\mathrm{higher})\;+\beta_{9}\;(\mathrm{employed}\;\text{full-time})\;+\beta_{10}\;(\mathrm{income})$$

In this equation, _i_ represents 1165 survey responses.

To explore differences across gender and race in the multilinear model, 6 additional models included interaction terms between cognitive factors (i.e., coronavirus crisis perception, perceived economic risk of coronavirus, general self-efficacy) and coronavirus anxiety. Predicted probabilities of coronavirus anxiety by cognitive factor were calculated and graphed among significant interactions to explore the moderation effect of gender and race. A one-tailed *p*-values of 0.05 or less were considered significant for all interaction model tests.

### 2.4. Ethical Considerations

The primary study was designed and conducted in accordance with the Declaration of Helsinki for research involving human subjects. The Ethics Review Panel (ERP) at the Université of Luxembourg approved the primary study protocol and procedures. Digital informed consent was obtained by participants prior to the start of the online survey from the primary study. This study was a secondary analysis of data collected in the primary study and deemed exempt by the Univeristy of California, Irvine Institutional Review Board.

## 3. Results

### 3.1. Demographics

The average age of survey respondent was 45.6 ± 15.7 years, and the majority identified as non-Hispanic White (76%). Respondents in this sample were highly educated, with 53% attaining a bachelor’s degree or above. Approximately 32% of participants were employed full-time at the time of the study. Females and non-White participants reported higher coronavirus anxiety. Table 1 presents additional sample characteristics of respondents and bivariate analyses with the dependent variable. Each independent variable—coronavirus crisis perception, perceived economic risk of coronavirus, and general self-efficacy—were significantly associated with coronavirus anxiety in bivariate analyses. Crisis perception and perceived economic risk were positively associated, whereas general self-efficacy was negatively associated with coronavirus anxiety. See Table 1.

### 3.2. Multiple Regression Model

The single multiple regression model adjusting for six covariates supported all hypotheses (Table 2). The first hypothesis (higher coronavirus crisis perception will be associated with higher coronavirus anxiety) was confirmed. Coronavirus crisis perception was positively associated with coronavirus anxiety after controlling for covariates. For every 1-unit increase in coronavirus crisis perception, there was a 0.46 increase in coronavirus anxiety (β = 0.46, 95% CI = 0.41, 1.00). The second hypothesis (higher economic risk perception will be associated with higher coronavirus anxiety) was also confirmed by the analyses. Perceived economic risk of coronavirus was positively associated with coronavirus anxiety. Controlling for age, gender, race, education, employment, and income, for every 1-unit increase in perceived economic risk of coronavirus, there was a 0.14 mean increase in coronavirus anxiety (β = 0.14, 95% CI = 0.09, 1.00). Finally, the third hypothesis (lower general self-efficacy will be associated with higher coronavirus anxiety) was confirmed (β = −0.15, 95% CI = −1.00, −0.10). The mean-adjusted *R*-square value indicated that the covariates explained approximately 32% of the variance in coronavirus anxiety.

### 3.3. Interaction Models

Three interaction models were run to evaluate the effect of gender on the association between each cognitive factor (i.e., coronavirus crisis perception, perceived economic risk of coronavirus, general self-efficacy) and coronavirus anxiety. The first interaction model showed statistical significance between females and males, indicating the slope of the relationship between coronavirus crisis perception and coronavirus anxiety differed by gender (β = 0.11, 95% CI = 0.03, 1.00). In the second and third models, gender was not a statistically significant moderator between the association of economic risk perception (β = 0.04, 95% CI = −0.04, 1.00) or self-efficacy (β = −0.06, 95% CI = −1.00, 0.03) on coronavirus anxiety.

Next, three interaction models were run to evaluate the effect of race on the association between each cognitive factor (i.e., coronavirus crisis perception, perceived economic risk of coronavirus, general self-efficacy) and coronavirus anxiety. The first interaction model showed statistical significance between White and non-White participants, indicating a difference in the relationship between coronavirus crisis perception and coronavirus anxiety by race (β = 0.17, 95% CI = 0.03, 1.00). In the second model, race also moderated the association between economic risk perception (β = 0.11, 95% CI = 0.01, 1.00) and coronavirus anxiety. Race was not a statistically significant moderator between self-efficacy and coronavirus anxiety (β = 0.15, 95% CI = −1.00, 0.12).

The results of significant interaction models are displayed in Table 3. Predicted probabilities of significant interaction models are presented in Figure 1, Figure 2 and Figure 3 to clarify interaction effects. The size of the effect of coronavirus crisis perception and coronavirus anxiety was greater for females and Whites. The size of the effect of economic risk perception on coronavirus anxiety was greater for Whites.

## 4. Discussion

This study explores various cognitive factors associated with coronavirus anxiety among subgroups disproportionately affected by anxiety in the general U.S. population. Using data from the start of the pandemic, our results indicate coronavirus crisis perception and perceived economic risk of coronavirus were positively associated with coronavirus anxiety and general self-efficacy was negatively associated with coronavirus anxiety when controlling for age, gender, race, education, employment, and income. Female gender and White race moderated the association between coronavirus crisis perception and coronavirus anxiety. Similarly, White race also moderated the association between economic crisis perception and coronavirus anxiety.

These findings support the RPA framework by showing perceptions and efficacy are tied to health outcomes such as anxiety. Using the RPA framework to simultaneously measure risk perception and self-efficacy, standardized coefficients suggest crisis risk perception carries a stronger association with coronavirus anxiety compared to economic risk and self-efficacy. To potentially alleviate coronavirus anxiety among the general population, coronavirus health messages should address how protective health behaviors reduce the harmful impacts of pandemics on a global level.

The coronavirus crisis perception measure used as an independent variable in this study was a more homogenous measure focused on a subjective assessment of the virus on a global scale, in contrast to prior risk-perception scales evaluating the SARS pandemic [25]. A cross-sectional study among health workers in China found risk perception of SARS to be positively associated with mental health outcomes such as PTSD [41]. Expanding the focus to the general population, this study found similar results supporting an association between coronavirus crisis perception and coronavirus anxiety in the United States. It is important to note increased coronavirus crisis perception may play an important role in individuals adopting preventive health behaviors (e.g., mask wearing) [42]. Future studies should simultaneously explore the role various perceptions have on positive health behaviors (e.g., mask wearing) and on negative mental health outcomes. Public health messaging can benefit from a better understanding of risk perceptions that influence health behaviors and do not contribute to negative mental health outcomes. This study verified that individual coronavirus crisis perception is an important factor and is associated with increased coronavirus anxiety. People with dysfunctional coronavirus anxiety can have debilitating psychological difficulties. This poses a major risk factor for psychopathology that disrupts daily life functioning [43]. Furthermore, as the COVID-19 pandemic continues, mental distress will likely continue and needs to be monitored [44]. Longitudinal trends in the association between coronavirus crisis perception and coronavirus anxiety should also be explored. A recent study found individual perceived risk of coronavirus increased from March to April 2020 [31]. A longitudinal study also found the emotion on social media posts during the COVID-19 pandemic increased over time, with high posting activity during the time frame of the current study [45]. The current study uncovers important predictors of anxiety at the start of the pandemic. Public health messaging that increases awareness of risk should include actionable recommendations to address the risk, also known as response efficacy messages. According to the extended parallel process communication model, anxiety-arousing risk messages need to be coupled with response efficacy messages to address anxiety-inducing coronavirus perceptions and avoid dysfunctional anxiety arousal [46,47]. This study identified the association between crisis risk perception and coronavirus anxiety differs by gender and race. Females have shown higher levels of anxiety and risk perception throughout the coronavirus pandemic [23]. Study findings add to this literature by identifying how these gender differences effect the association between crisis risk perception and coronavirus anxiety compared to male counterparts. Messaging that addresses actions to evade a global crisis may ease anxiety among all genders but may be an opportunity to ease anxiety among females which have reported higher levels of anxiety throughout the pandemic.

Studies have documented the association between financial and economic hardship and mental health outcomes, including those related to the coronavirus pandemic [26,48,49,50]. This study confirmed the association between perceived economic risk of coronavirus and coronavirus anxiety in the U.S. general population 1 month after the federal stay-at-home order was announced, just before the economy began to feel the effects of the pandemic. The perceived economic risk of coronavirus and its relationship with increased anxiety may have continued, with estimated true unemployment at 26% and higher among minority subgroups (e.g., 50.2% among Black Americans) [51]. A previous China-based study found males reported higher economic and work-related insecurity [52]. Results of the current U.S.-based study found gender did not moderate the effect of economic risk perception and anxiety. The effect of gender on perceived economic risk of coronavirus among individuals of different groups job sectors and socioeconomic classes should be evaluated. The current study supports previous work that shows non-White individuals were adversely affected by coronavirus anxiety [16,17], specifically economic anxiety [17]. Our results indicate the slope of the association between economic risk perception and coronavirus anxiety was steeper for White individuals compared to non-Whites. Longitudinal studies exploring racial disparities of economic-related anxiety may be able to better confirm associations between race, economic risk and anxiety. It is possible that disparities in economic-related anxiety were temporal and fluctuated throughout the course of the pandemic.

Growing empirical support suggests self-efficacy may buffer some of the negative effects of mental health [53]. Previous work has documented the association between lower self-efficacy and anxiety among health workers during the coronavirus pandemic [28,29]. The current study found lower general self-efficacy to be significantly associated higher coronavirus anxiety. At the time of the survey (13–15 March 2020), many coronavirus transmission pathways and prevention recommendations were not well-researched and messaging regarding protective prevention measures had not yet been disseminated to the general public. It is possible that the association between general self-efficacy and coronavirus anxiety strengthened during the pandemic as awareness and understanding of actionable behaviors to minimize the risk of infection increased. Additional studies are needed to test the association between general self-efficacy and coronavirus anxiety as prevention recommendations changed with emerging science.

Strengths of this study include early reporting on the outcome of coronavirus-specific anxiety measures that captured a composite of cognitive and somatic dimensions. This study builds on prior coronavirus studies [28,29,42] by examining the association between self-efficacy and anxiety by using a coronavirus anxiety measure that includes both cognitive and somatic items. Additionally, this study used participants from Prolific, a dedicated respondent pool for academic research. Advantages of the Prolific survey population include a wide reach and quick results from respondents who are relatively more naïve compared with other respondent populations such as MTurk [54], with less experienced survey takers completing surveys.

### Limitations

Generalizations from the study need to be tempered by the fact that the Prolific online survey respondent population presents a bias toward female, young, highly educated, and White participants. In addition, respondents chose to participate, indicating they may be interested in the topic and may be systematically different from other populations [55]. Similar to limitations of other online surveys [56,57], sample characteristics with Prolific are self-reported and the platform is limited to those with online access, tends to have more politically and civically engaged participants, and more self-reported Democratic participants. Interpretation of results need to be tempered by reported respondent characteristics. Findings from this study are more generalizable to participants of similar characteristics. Of survey participants who completed this survey, more than half (54%) self-reported as Democratic [58]. In addition, this study included a small national sample of 1165 participants. A second limitation relates to the fact that no state-level data were collected for the mid-March survey; therefore, regional U.S. differences in coronavirus crisis perception could not be captured or accounted for in the analysis. A third limitation is that this paper used a Coronavirus anxiety measure that does not evaluate the severity of anxiety symptoms. Additionally, perceived economic risk of coronavirus was measured with only one item. Only 32% of coronavirus anxiety was explained by the independent variables in the model. Still, more work can be conducted to investigate other factors that influenced coronavirus anxiety at the early stages of the pandemic. It is also possible that crisis risk perception, economic risk perception and self-efficacy explained a higher percentage of coronavirus anxiety at later stages of the pandemic. Data were obtained within a short time frame. The cross-sectional nature of this study allows us to only report on associations at the early stages of the pandemic and findings may have ceased or exacerbated as the pandemic progressed. Additionally, survey implementation on 13 March coincided with President Donald Trump declaring a national emergency due to COVID-19, which may have contributed to exacerbated coronavirus crisis perceptions among the survey respondents.

## 5. Conclusions

This study provides initial insight into coronavirus anxiety in the general U.S. population. After controlling for covariates, this study found associations between higher coronavirus crisis perception and higher coronavirus anxiety, higher perceived economic risk and higher coronavirus anxiety, and lower self-efficacy and higher coronavirus anxiety. Our study has practical implications to psychiatry. The World Psychiatric Association outlines the role of psychiatrists in advocating for government interventions that reduce stress and suicide in the general population [59]. Our study suggests coronavirus crisis perception and perceived economic risk of coronavirus may be valuable factors to consider in public health messaging when addressing global health crises and associated anxiety. Further, psychiatrists should prepare for a persistent demand in mental health services long after the country financially recovers from economic implications of COVID-19. Previous periods of economic challenges have shown the mental health impact of long-term employment after a country’s financial recovery [60]. Future studies replicating these findings will build confidence regarding the importance of the relationship between pandemic risk perception, perceived economic risk of coronavirus, and increased anxiety among individuals with higher levels of coronavirus anxiety. Researchers should evaluate cognitive factors that attribute to positive health behaviors and negative mental health outcomes among subgroups disproportionately affected by anxiety in order to better inform public health messaging.

## Figures and Tables

**Figure 1 ijerph-19-02872-f001:**
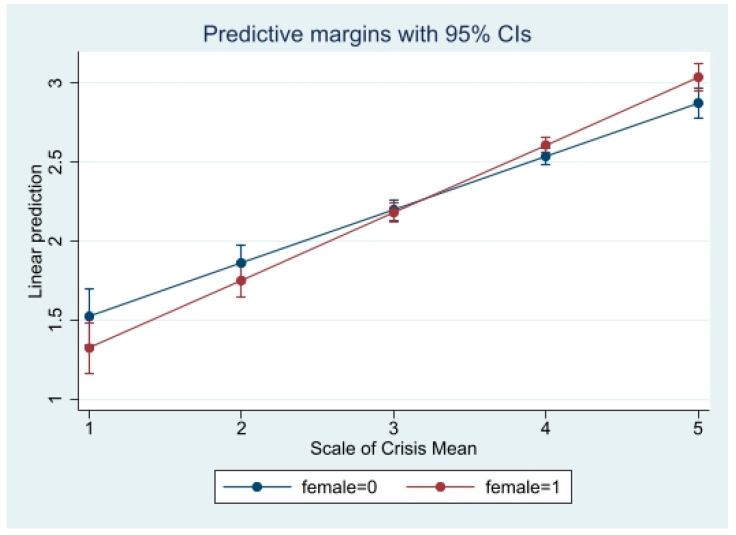
Predicted Coronavirus Anxiety by Crisis Perception and Gender.

**Figure 2 ijerph-19-02872-f002:**
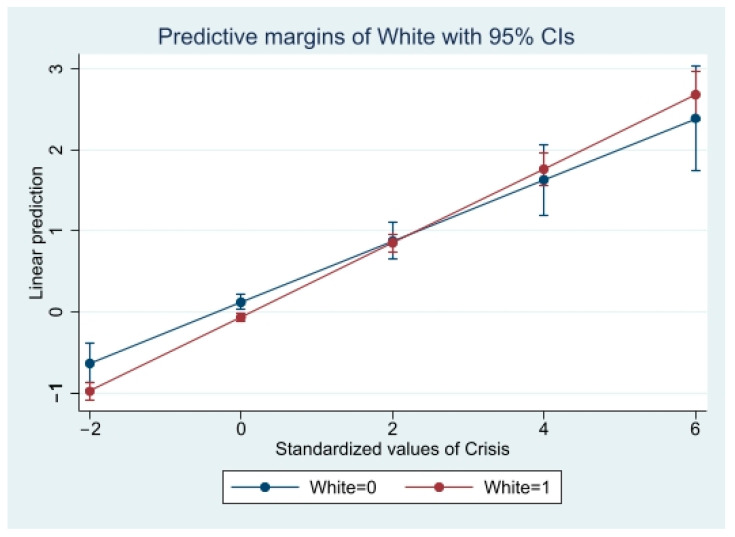
Predicted Coronavirus Anxiety by Crisis Perception and Race.

**Figure 3 ijerph-19-02872-f003:**
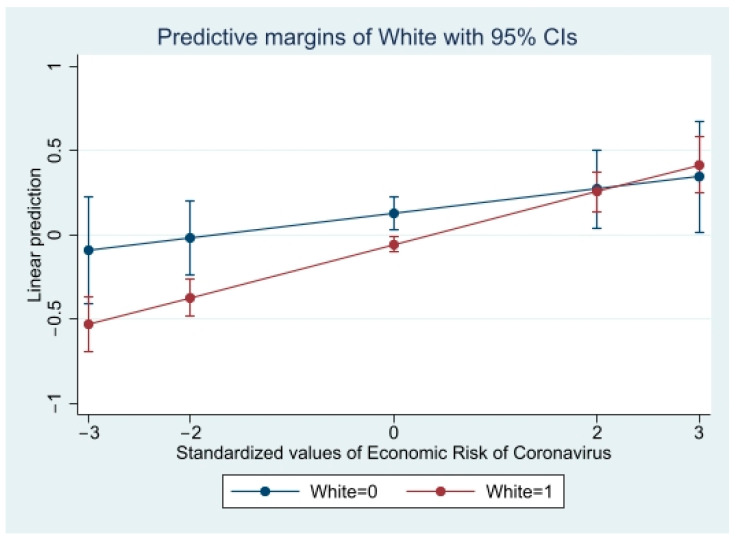
Predicted Coronavirus Anxiety by Perceived Economic Risk of Coronavirus and Race.

**Table 1 ijerph-19-02872-t001:** Descriptive Characteristics of Participants by Coronavirus Anxiety.

Variable	Sample Total (*N* = 1182)	β (*p*)
Coronavirus Anxiety	2.49 ± 0.66	-
Coronavirus crisis perception, ^a,b^ mean ± *SD*	3.82 ± 0.83	0.42 (<0.001)
Perceived economic risk of Coronavirus, ^a,c^ mean ± *SD*	3.89 ± 0.99	0.23 (<0.001)
General self-efficacy, ^a,d^ mean ± *SD*	3.71 ± 0.72	−0.12 (<0.001)
Gender, ^e^ *n* (%)		
Female ^f^	596 (50.42)	0.07 (0.04)
Male ^f^	574 (48.56)	−0.08 (0.01)
Other ^f^	12 (1.02)	0.02 (0.43)
Race, ^g^ *n* (%)		−0.16 (<0.001)
African American	141 (11.93)	
Asian	64 (5.41)	
Hispanic	47 (3.98)	
Native American	7 (0.59)	
Non-Hispanic White	903 (76.40)	
Other	20 (1.69)	
Annual Income, ^e,h^ *n* (%)		−0.01 (0.42)
No personal income	30 (2.54)	
Less than USD 20,000	179 (15.17)	
USD 20,000 to USD 34,999	190 (16.07)	
USD 35,000 to USD 49,999	190 (16.07)	
USD 50,000 to USD 74,999	215 (18.19)	
USD 75,000 to USD 99,999	144 (12.18)	
USD 100,000 up to USD 114,999	77 (6.51)	
USD 115,000 up to USD 129,999	45 (3.81)	
USD 130,000 or more	95 (8.04)	
Do not wish to respond	17 (1.44)	
Education level, ^f,i^ *n* (%)		−0.01 (0.80)
Less than a high school diploma	6 (0.51)	
High school diploma or equivalent	124 (10.49)	
Some college, no degree	269 (22.76)	
Associate’s degree	139 (11.76)	
Bachelor’s degree	414 (35.03)	
Master’s degree	155 (13.11)	
Professional degree	33 (2.79)	
Doctorate degree	33 (2.79)	
Vocational training/trade	7 (0.59)	
Other; specify	2 (0.17)	
Age (years), ^f^ mean ± *SD*	45.6 ± 15.7	−0.001 (0.08)
Employment, ^f,j^ *n* (%)		−0.03 (0.38)
Employed full time (≥40 h/week)	428 (36.21)	
Employed part time (<40 h/week)	157 (13.28)	
Unemployed and currently looking for work	78 (6.60)	
Unemployed and currently not looking for work	12 (1.02)	
Student	61 (5.16)	
Retired	176 (14.89)	
Homemaker	63 (5.33)	
Self-employed	161 (13.62)	
Unable to work	46 (3.89)	

^a^ Results reflect one-tailed significance test of bivariate associations from Simple Linear Regression tests with Coronavirus anxiety as the dependent variable. ^b^ Coronavirus Crisis Perception Scale [39], with 5 items and responses ranging from 1 (Strongly Disagree) to 5 (Strongly Agree). Mean scores were computed. Higher scores reflect higher crisis perception. ^c^ Participants were asked to respond to the statement, “I think Coronavirus will be a disaster for our economy”. Responses ranging from 1 (Strongly Disagree) to 5 (Strongly Agree). Higher scores reflect higher perceived economic risk of Coronavirus. ^d^ New General Self-Efficacy Scale [38], with 8 items and responses ranging from 1 (Strongly Disagree) to 5 (Strongly Agree). Mean scores were computed. Higher scores reflect higher self-efficacy. ^e^ Gender responses for gender were dummy-coded. ^f^ Results reflect two-tailed significance test of bivariate associations from ANOVA and Simple Linear Regression tests with Coronavirus anxiety as the dependent variable. ^g^ Race categories were recoded as (0) Non-White and (1) White. ^h^ Income categories were coded as continuous with ‘do not wish to respond’ treated as missing. ^i^ Responses for educational status were recoded into (0) Associate’s degree or less, and (1) Bachelor’s degree or higher. ^j^ Responses were recoded as (1) Employed full time (40) or more hours a week, and (0) other.

**Table 2 ijerph-19-02872-t002:** Associations Between Cognitive Factors and Coronavirus Anxiety.

	Coronavirus Anxiety ^ab^	
	β (95% CI)	*p*-Value
Coronavirus crisis perception ^c^	0.46 (0.41, 1.00)	<0.001
Perceived economic crisis of Coronavirus ^d^	0.14 (0.09, 1.00)	<0.001
Self-efficacy ^e^	−0.15 (−1.00, −0.11)	<0.001

Note. Results reflect standardized beta coefficients and one-tailed significance test values from multivariate regression. ^a^ The Coronavirus Anxiety Scale [33], with 5 items and responses ranging from 1 (Strongly Disagree) to 5 (Strongly Agree). Three items were reverse coded. Mean scores were computed. Higher scores indicated higher levels of dispositional mindfulness. ^b^ Model of the association between cognitive factors (i.e., Coronavirus crisis perception, perceived economic risk of Coronavirus, general self-efficacy) and coronavirus anxiety was adjusted for age, gender, race, education level, employment status, and income. ^c^ Coronavirus Crisis Perception Scale [39] consists of 5 items with responses ranging from 1 (Strongly Disagree) to 5 (Strongly Agree). Mean scores were computed. Higher scores reflect higher crisis perception. ^d^ Participants were asked to respond to the statement, “I think Coronavirus will be a disaster for our economy”. Responses ranging from 1 (Strongly Disagree) to 5 (Strongly Agree). Higher scores reflect higher perceived economic risk of Coronavirus. ^e^ New General Self-Efficacy Scale [38], with 8 items and responses ranging from 1 (Strongly Disagree) to 5 (Strongly Agree). Mean scores were computed. Higher scores reflect higher self-efficacy.

**Table 3 ijerph-19-02872-t003:** Associations Between Cognitive Factors and Coronavirus Anxiety by Gender and Race.

	Coronavirus Anxiety ^ab^	
	β (95% CI)	*p*-Value
Interaction Model by Gender		
Coronavirus crisis perception ^c^	0.40 (0.34, 1.00)	<0.001
Perceived economic crisis of Coronavirus ^d^	0.14 (0.10, 1.00)	<0.001
Self-efficacy ^e^	−0.15 (−1.00, −0.10)	<0.001
Coronavirus crisis perception ^c^ * gender (female = ref)	0.11 (0.03, 1.00)	0.01
Interaction Model by Race		
Coronavirus crisis perception ^c^	0.32 (0.19, 1.00)	<0.001
Perceived economic crisis of Coronavirus ^d^	0.14 (0.09, 1.00)	<0.001
Self-efficacy ^e^	−0.15 (−1.00, −0.11)	<0.001
Coronavirus crisis perception ^c^ * race (White = ref)	0.17 (0.03, 1.00)	0.02
Interaction Model by Race		
Coronavirus crisis perception ^c^	0.46 (0.41, 1.00)	<0.001
Perceived economic crisis of Coronavirus ^d^	0.05 (−0.04, 1.00)	0.35
Self-efficacy ^e^	−0.16 (−1.00, −0.10)	<0.001
Perceived economic crisis of Coronavirus ^d^ * race (White = ref)	0.11 (0.01, 1.00)	0.03

Note. Results reflect standardized beta coefficients and one-tailed significance test values from multivariate regression. ^a^ The Coronavirus Anxiety Scale [33], with 5 items and responses ranging from 1 (Strongly Disagree) to 5 (Strongly Agree). Three items were reverse coded. Mean scores were computed. Higher scores indicated higher levels of dispositional mindfulness. ^b^ Model of the association between cognitive factors (i.e., Coronavirus crisis perception, perceived economic risk of Coronavirus, general self-efficacy) and coronavirus anxiety was adjusted for age, gender, race, education level, employment status, and income. ^c^ Coronavirus Crisis Perception Scale [39] consists of 5 items with responses ranging from 1 (Strongly Disagree) to 5 (Strongly Agree). Mean scores were computed. Higher scores reflect higher crisis perception. ^d^ Participants were asked to respond to the statement, “I think Coronavirus will be a disaster for our economy”. Responses ranging from 1 (Strongly Disagree) to 5 (Strongly Agree). Higher scores reflect higher perceived economic risk of Coronavirus. ^e^ New General Self-Efficacy Scale [38], with 8 items and responses ranging from 1 (Strongly Disagree) to 5 (Strongly Agree). Mean scores were computed. Higher scores reflect higher self-efficacy. Asterisks in this table reflect interactions between to variables.

## Data Availability

Additional information on measures and findings may be found here: https://osf.io/dkv2s/?view_only=e522a25ac66d484e9afaa2644af25e89 (accessed on 8 October 2021).

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
