# Peer review of "Associations between Coronavirus Crisis Perception, Perceived Economic Risk of Coronavirus, General Self-Efficacy, and Coronavirus Anxiety at the Start of the Pandemic: Differences by Gender and Race"

_ijerph, 2022, doi:10.3390/ijerph19052872_

Round 1
Reviewer 1 Report
The study used data from an online survey to examine associations between coronavirus crisis perception, perceived economic risk of coronavirus, general self-efficacy, and coronavirus anxiety in the us population. The major concern is the lack of a clear conceptual framework for the studied associations and how this study can add to the literature.
Since the pandemic outbreak, numerous studies have examined a wide range of factors associated with population anxiety and other mental health symptoms, including the concerning factors in the current study. These existing studies should be introduced more inclusively, but they also impose a question about what the current study can add to the literature.
A second major concern is the lack of a clear theoretical framework underlying the examined associations across factors. It is unclear why these specific factors were highlighted as predictors of anxiety in the study. Given that many studies have examined these factors’ association with anxiety during the pandemic, it may be more worthwhile to advance the exploration by investigating more structural associations among them, such as mediating or moderating associations, if there is a theoretical framework to conceptualize such associations.
The paper mentioned that missing data were handled using multiple imputation by chained equations with 20 iterations. Data were imputed for income with < 2% of missing data (line 145-146). Does that mean the analyses were done with 20 imputed datasets and then the results were synthesized? When conducting multiple imputation, aren’t all variables with missing values imputed? The authors may want to provide some more specifics.
The authors may want to interpret the findings in a more updated literature context, given that numerous studies related to the topic have been done. For example, the authors suggested that future studies should evaluate how the perceived economic risk of coronavirus has changed in the following months and explore groups adversely affected by economic circumstances. Are there already studies in these aspects? If so, it is better to discuss the findings in a more updated context.
Author Response
Comment #1: The study used data from an online survey to examine associations between coronavirus crisis perception, perceived economic risk of coronavirus, general self-efficacy, and coronavirus anxiety in the us population. The major concern is the lack of a clear conceptual framework for the studied associations and how this study can add to the literature.
Response #1: The authors agree with this comment. Per the suggestion in comment #3, the authors now explore moderation effects among subgroups most adversely impacted by anxiety throughout the coronavirus pandemic. In addition, the authors now use the risk perception attitude framework which suggests risk perceptions impact on health behaviors and outcomes are best explored in the context of self-efficacy. By measuring self-efficacy alongside crisis risk perception and economic risk perception, this study is able to better suited to evaluate the impact of risk perception on anxiety. Major revisions to our theoretical framework can be found on lines 66-75.
Comment #2: Since the pandemic outbreak, numerous studies have examined a wide range of factors associated with population anxiety and other mental health symptoms, including the concerning factors in the current study. These existing studies should be introduced more inclusively, but they also impose a question about what the current study can add to the literature.
Response #2: The authors agree addition studies that explore anxiety throughout the pandemic should be addressed more inclusively throughout the paper. Based on suggestions provided in comments 3 and 5, we believe the paper can add to the literature by further exploring moderators to assess if and how individuals at higher risk of coronavirus anxiety throughout the pandemic (e.g., females and racial/ethnic minorities) differed in their association between cognitive factors and coronavirus anxiety. Perhaps these moderations may provide insight into disparities observed early on at the start of the pandemic.
Comment #3: A second major concern is the lack of a clear theoretical framework underlying the examined associations across factors. It is unclear why these specific factors were highlighted as predictors of anxiety in the study. Given that many studies have examined these factors’ association with anxiety during the pandemic, it may be more worthwhile to advance the exploration by investigating more structural associations among them, such as mediating or moderating associations, if there is a theoretical framework to conceptualize such associations.
Response #3: The authors appreciate the suggestion to present a clearer theoretical framework and to further explore potential mediating or moderating associations. The authors feel moderation investigation with this cross-sectional data is most appropriate and can shed light on how disadvantages groups differ in anxiety. Given the unique time point of when this data is collected, the authors believe a deeper investigation of how the association between cognitive factors and coronavirus anxiety may differ by race and gender. Literature has shown females and minority groups may be more susceptible to anxiety and mental health issues during the Coronavirus pandemic. Understanding if and how gender and race moderated the associations between crisis risk perception, self-efficacy, economic risk perception and coronavirus anxiety at the start of the pandemic may elucidate how these disparities evolved throughout the course of the pandemic.
Comment #4: The paper mentioned that missing data were handled using multiple imputation by chained equations with 20 iterations. Data were imputed for income with < 2% of missing data (line 145-146). Does that mean the analyses were done with 20 imputed datasets and then the results were synthesized? When conducting multiple imputation, aren’t all variables with missing values imputed? The authors may want to provide some more specifics.
Response #4: We thank reviewer 1 for this comment. Multiple imputation by chained equations (MICE) is an iterative process where missing values are replaced with imputed (predictions) for specified number of cycles (e.g., 20) improving and updating the estimates with every iteration.1 Some literature supports running 20 iterations to produce a stable model where parameters converge.2 The authors have discussed this comment and believe case-wise deletion is a better approach for this data. MICE is unnecessary with only one variable and other literature suggests imputation may be inappropriate for such a low threshold.3 The statistical methods section has been updated to reflect current changes discussed.
- Azur, M.J., Stuart, E.A., Frangakis, C. and Leaf, P.J., 2011. Multiple imputation by chained equations: what is it and how does it work?. International journal of methods in psychiatric research, 20(1), pp.40-49.
- White, I.R., Royston, P. and Wood, A.M., 2011. Multiple imputation using chained equations: issues and guidance for practice. Statistics in medicine, 30(4), pp.377-399.
- Tabachnick, B.G., Fidell, L.S. and Ullman, J.B., 2007. Using multivariate statistics(Vol. 5, pp. 481-498). Boston, MA: Pearson.
Comment #5: The authors may want to interpret the findings in a more updated literature context, given that numerous studies related to the topic have been done. For example, the authors suggested that future studies should evaluate how the perceived economic risk of coronavirus has changed in the following months and explore groups adversely affected by economic circumstances. Are there already studies in these aspects? If so, it is better to discuss the findings in a more updated context.
Comment #5: We thank reviewer 1 for this suggestion. Building on this suggestion and on the suggestion in comment #3, the authors now explore groups (i.e., females and racial/ethnic minorities) more adversely affected by anxiety throughout the pandemic. The authors made major revisions to the introduction and discussion sections to more inclusively discuss past work, what our findings contribute, and necessary next steps.
Reviewer 2 Report
I really dont think this study should be published. The data is collected within four days and that was the begining of the coronavirus pandemic where people are generally learning about the disease. Hence it is not clear how the data can be used to measure the anxiety of US population.
Most absurd thing about the study is data collection period was March 15 and 20 when the infection was just started to spread. This is not acceptable at all.
This paper should be rejected at the desk level.
Author Response
Comment #1: I really dont think this study should be published. The data is collected within four days and that was the begining of the coronavirus pandemic where people are generally learning about the disease. Hence it is not clear how the data can be used to measure the anxiety of US population.
Most absurd thing about the study is data collection period was March 15 and 20 when the infection was just started to spread. This is not acceptable at all.
This paper should be rejected at the desk level.
Response #1: We thank reviewer 2 for this comment and the chance to elaborate on why the data collection period could be viewed as a strength. This is a cross-sectional study with data collected at the very start of a historical moment where the first time in US history, the country experienced business and school shutdowns across the nation. While cross-sectional studies are limited to inferences that can be made by longitudinal datasets, cross-sectional data offers the ability to provide information in a single point in time. In this study, details on American anxiety at the early start of the pandemic may provide insight on how health messages that can be better tailored to address early anxiety of the population during global pandemics. In addition, cross-sectional studies can provide important foundation for longitudinal investigations later on. While the authors agree there are limitations to the inferences that can be made with cross-sectional data, the authors disagree with Reviewer 2’s assumption that data collected in this time point is not of value.
Reviewer 3 Report
Dear authors and editor,
The manuscript titled "Associations between Coronavirus Crisis Perception, Perceived Economic Risk of Coronavirus, General Self-Efficacy, and Coronavirus Anxiety" This is a descriptive cross-sectional study that examine coronavirus anxiety as the dependent variable, alongside three independent variables: coronavirus crisis perception, perceived economic risk of coronavirus, and general self-efficacy.
There are many minor and major issues I'd like the authors resolve.
Abstract
1-adequate
2- Change the keywords. Delete the words "perceived risk " "crisis" and "economic risk.
Introduction
3-adequate
Materials and Methods
4-It is recommended to include a code of ethics.
5-Study size: Explain how the study size was arrived at. It is recommended to explain the sample size of descriptive studies in order to know the scope of the study.
Results
6-adequate
Discussion
7-It is recommended that the study design and sample size be added as a limitation. Only 31% of the dependent variable is explained. Line 208-209 "The mean-adjusted R-square value indicated that the covariates explained approximately 31% of the variance in coronavirus anxiety."
Conclusion
8-adequate
Reference:
9-adequate
Author Response
Abstract
Comment #1: Change the keywords. Delete the words "perceived risk " "crisis" and "economic risk.
Response #1: The words "perceived risk " "crisis" and "economic risk” have been removed.
Materials and Methods
Comment #2: It is recommended to include a code of ethics.
Response #2: We thank reviewer 3 for this recommendation. An ethical consideration section has been added with section 2.4.
Comment #3: Study size: Explain how the study size was arrived at. It is recommended to explain the sample size of descriptive studies in order to know the scope of the study.
Response #3: We thank the reviewer for this recommendation. An additional paragraph has been added to section 2.1 Study Sample explaining the sample size motivation. In addition, we further elaborated on the previous usage of the dataset in other collaborations.
Discussion
Comment #4: It is recommended that the study design and sample size be added as a limitation. Only 31% of the dependent variable is explained. Line 208-209 "The mean-adjusted R-square value indicated that the covariates explained approximately 31% of the variance in coronavirus anxiety."
Response #4: We thank reviewer 3 for this comment. The limitation section of the discussion now includes the small sample size and the model explained only 31% of coronavirus anxiety
Round 2
Reviewer 1 Report
The authors generally addressed my concerns in the comments. The authors may consider providing some information about the pandemic scenario in the US at the time of data collection for a better understanding of the study context. In addition, in the introduction or discussion, it would be helpful to refer to studies that examined the dynamics of anxiety/mental heath after the pandemic outbreak to show the value of advancing the understanding of the issue at the beginning stage.
Zhang, S., Liu, M., Li, Y., & Chung, J. E. (2021). Teens’ Social Media Engagement during the COVID-19 Pandemic: A Time Series Examination of Posting and Emotion on Reddit. International Journal of Environmental Research and Public Health, 18(19), 10079.
Yarrington, J. S., Lasser, J., Garcia, D., Vargas, J. H., Couto, D. D., Marafon, T., ... & Niles, A. N. (2021). Impact of the COVID-19 pandemic on mental health among 157,213 Americans. Journal of Affective Disorders, 286, 64-70.
Author Response
Comment #1: The authors generally addressed my concerns in the comments. The authors may consider providing some information about the pandemic scenario in the US at the time of data collection for a better understanding of the study context.
Response #1: We thank Reviewer 1 for this suggestion. The introduction now provides additional insight into the state of the US in midst of the pandemic during the time of data collection.
Comment #2: In addition, in the introduction or discussion, it would be helpful to refer to studies that examined the dynamics of anxiety/mental heath after the pandemic outbreak to show the value of advancing the understanding of the issue at the beginning stage.
- Zhang, S., Liu, M., Li, Y., & Chung, J. E. (2021). Teens’ Social Media Engagement during the COVID-19 Pandemic: A Time Series Examination of Posting and Emotion on Reddit. International Journal of Environmental Research and Public Health, 18(19), 10079.
- Yarrington, J. S., Lasser, J., Garcia, D., Vargas, J. H., Couto, D. D., Marafon, T., ... & Niles, A. N. (2021). Impact of the COVID-19 pandemic on mental health among 157,213 Americans. Journal of Affective Disorders, 286, 64-70.
Response #2: We thank reviewer 1 for sharing these helpful papers. The authors have incorporated Zhang et al. 2021 into the discussion and Yarrington et al. 2021 into the introduction.
Reviewer 2 Report
Should be rejected
Author Response
Comment #1: Should be rejected.
Response #1: No constructive feedback was provided or rationale was provided by reviewer 2. The authors maintain this paper contributes to existing literature by highlighting anxiety predictors at the start of the pandemic.
Reviewer 3 Report
Dears authors;
I am satified about revised version, it is suitable to be accepted now.
Kind regards.
Author Response
Comment #1: I am satisfied about revised version, it is suitable to be accepted now.
Response #1: The authors thank reviewer 3 for helpful comments during the first round of revisions.